# Deficits in Prenatal Serine Biosynthesis Underlie the Mitochondrial Dysfunction Associated with the Autism-Linked *FMR1* Gene

**DOI:** 10.3390/ijms22115886

**Published:** 2021-05-30

**Authors:** Sarah L. Nolin, Eleonora Napoli, Amanda Flores, Randi J. Hagerman, Cecilia Giulivi

**Affiliations:** 1Department of Human Genetics, New York State Institute for Basic Research in Developmental Disabilities, Staten Island, NY 10314, USA; sallynolin@gmail.com; 2Department of Molecular Biosciences, School of Veterinary Medicine, University of California, Davis, CA 95616, USA; enapoli@ucdavis.edu (E.N.); accflores@ucdavis.edu (A.F.); 3Medical Sciences Campus, Department of Biochemistry, University of Puerto Rico, San Juan PR00936, Puerto Rico; 4Department of Pediatrics, University of California Davis Medical Center, Sacramento, CA 95817, USA; rjhagerman@ucdavis.edu; 5The MIND Institute, University of California Davis Medical Center, Sacramento, CA 95817, USA

**Keywords:** amniotic fluid, premutation, CGG repeats, *FMR1*, metabolomics, proteomics

## Abstract

Fifty-five to two hundred CGG repeats (called a premutation, or PM) in the 5′-UTR of the *FMR1* gene are generally unstable, often expanding to a full mutation (>200) in one generation through maternal inheritance, leading to fragile X syndrome, a condition associated with autism and other intellectual disabilities. To uncover the early mechanisms of pathogenesis, we performed metabolomics and proteomics on amniotic fluids from PM carriers, pregnant with male fetuses, who had undergone amniocentesis for fragile X prenatal diagnosis. The prenatal metabolic footprint identified mitochondrial deficits, which were further validated by using internal and external cohorts. Deficits in the anaplerosis of the Krebs cycle were noted at the level of serine biosynthesis, which was confirmed by rescuing the mitochondrial dysfunction in the carriers’ umbilical cord fibroblasts using alpha-ketoglutarate precursors. Maternal administration of serine and its precursors has the potential to decrease the risk of developing energy shortages associated with mitochondrial dysfunction and linked comorbidities.

## 1. Introduction

The fragile X mental retardation 1 (*FMR1*) gene contains CGG repeats that can expand to >200 copies (full mutation), causing fragile X syndrome, a condition frequently co-diagnosed with autism and other intellectual disabilities [1,2]. Normal alleles (10–44 repeats) are highly stable; however, transmissions of the premutation (PM; 55–200 repeats) may expand to full mutation in one generation through maternal inheritance [3].

Adult PM carriers are at higher risk of developing the late-onset fragile-X-associated tremor/ataxia syndrome (FXTAS), characterized by progressive intention tremors, gait ataxia, deficits in executive function and memory, peripheral neuropathy, and parkinsonism [4,5]. Children with the PM show higher incidence of attention deficit, aggressiveness, social anxiety, autism, and seizures [6,7]. In terms of developmental disabilities, a number of studies suggest a higher than expected rate of developmental delays in children and adolescents with the PM (see [6,7] and references herein), whereas others failed to identify such a link [8]. While an extensive analysis of the factors contributing to this discrepancy is outside the scope of this study, it is noteworthy that many of these reports may suffer from selection bias, as they were performed with clinically referred children. However, we cannot exclude the possibility that other variables (i.e., age at testing, CGG repeat size, mosaicism, CGG repeat structure) may play a role as relative risk factors for more or less subtle developmental differences observed in the PM pediatric population [7].

Mitochondrial dysfunction (MD) is at the center of many disorders, especially those affecting neurological systems, due to the heavy reliance of neurons on mitochondrial ATP [9]. In this context, our team reported for the first time MD in bio-samples from pediatric and adult carriers, with and without FXTAS [10,11], some of which were later replicated and confirmed by others [12,13,14]. More importantly, the MD observed in PBMCs was associated with CNS deficits (i.e., lower executive function and IQ, increased anxiety, fatigue, and attention deficit hyperactivity disorder [15]), and evidenced before the overt accumulation of nuclear inclusions—a hallmark of FXTAS [16]—or behavioral deficits [10]. These studies validated the use of peripheral markers as prognostic tools for the PM, consistent with other studies on neurological diseases [17,18,19]. This issue is of importance, as the identification of altered mitochondrial markers expressed in blood or other biofluids accessible without major invasive procedures [20,21], especially when testing children, should be valuable under the assumption that many such defects may not be confined only to tissues highly impacted by MD (such as brain tissues), opening the window for the use of peripheral biomarkers to allow early detection and monitor the potential response to therapeutic approaches.

As the *FMR1*-encoded fragile X mental retardation protein (FMRP) seems to have a strong neurodevelopmental role based on its high prenatal expression followed by its decline after birth in mice [22], the putative impact on metabolism that the PM may cause during prenatal periods remains unresolved. Here, we identified the prenatal metabolic footprint of the PM in amniotic fluid by utilizing a combined multi-omics approach. Pathway analyses allowed us to assess the impact of the PM during fetal growth—the most vulnerable period for brain development—followed by in vitro experiments that supported a critical role for serine biosynthesis in sustaining mitochondrial-derived energy.

## 2. Results & Discussion

### 2.1. Selection and Characterization of Amniotic Fluid Samples 

In order to identify the prenatal metabolic footprint of the PM, we examined the proteomic and metabolomic profiles of 54 amniotic fluid (AF) supernatants obtained from 52 pregnant carrier women who had undergone amniocentesis for prenatal fragile X diagnosis. All mothers carried a male fetus who had inherited either a PM or a non-mutated fragile X allele (Table 1). Due to the direct correlation between CGG repeat length and increased oxidative stress and mitochondrial dysfunction [22,23], only mothers with similar CGG expansions were selected. The maternal CGG repeats ranged—for the mutant allele—from 53 to 200 (mean ± SD: 72 ± 30) for mothers carrying a non-carrier fetus, and from 55 to 82 (mean ± SD: 63 ± 8) for those carrying a fetus with the PM, with no significant differences between the two groups (*t* (33) = −1.493; *p* = 0.143; *d* = 0.389; unequal variances *F* = 16.13; *p* < 0.0001). The age of the mothers carrying a non-carrier fetus ranged from 25.5 to 41.8 y (34 ± 4), slightly older than those carrying a PM fetus, who ranged from 17.8 to 42.1 y (30 ± 7; *t* (33) = −2.597; *p* = 0.0138; *d* = 0.751; unequal variances *F* = 2.334; *p* = 0.035). The fetal repeats, as expected, were significantly different between groups, ranging from 20 to 44 for the non-carrier group, and from 55 to 157 for the carrier group (mean ± SD: 29 ± 5 and 69 ± 21; *t* (23) = 9.908; *p* < 0.0001; *d* = 2.938; unequal variances *F* = 13.75; *p* < 0.0001). Most (88.9%; *p* < 0.0001) of the transmitted PM alleles were unstable (i.e., ± ≥1 repeat difference from the mother’s, as defined in [24]). As expected, most non-carrier fetuses presented two AGG interruptions in the CGG repeat stretch (65%), whereas most carriers exhibited fewer interruptions (68.2%) and of longer repeat size (Table 1). Moreover, while most of the non-carriers had a limited number of AGG interruption configurations (8 combinations), carriers exhibited a more varied pattern (15 combinations; Table 1).

Samples were taken from female carriers at gestational periods ranging from 13.3 to 21.2 weeks, with no statistical differences between the two groups (mean ± SD: 17 ± 2 gestational weeks and *n* = 30; 17 ± 1 and *n* = 22; *t* (50) = -0.838; *p* = 0.406; *d* = 0.228; Table 1). This gestational age was selected because the first 15–20 weeks are the most vulnerable for neurodevelopment [25], and the relatively narrow range minimizes putative metabolic profile differences that occur across the gestational period [26,27]. In support of this last premise, no correlation was observed between the fetal biomarker alpha-fetoprotein (AFP) and gestational age [28], as expected for the short timeframe (13–21 weeks) of sample collection, thereby allowing for a direct case–control comparison independent of the exact gestational week. Another argument that supports the selection of this interval is based on the fact that AF supernatants during these gestational weeks serve as suitable proxies for intra-fetal composition, mainly derived from fetal fluids. From 10 to 20 weeks of gestation, free diffusion occurs bi-directionally between AF and the fetus across the fetal skin, placenta, and umbilical cord, resulting in an AF composition similar to that of fetal plasma [29]. This is a direct consequence of the immature development of both keratinization and blood–brain barrier permeability [29], thereby allowing the passage of proteins between organs (including the brain) and the bloodstream [30].

### 2.2. Combined Omics as a Tool to Validate the Biological Matrix as Fetal Amniotic Fluid

As reported by other studies utilizing AF supernatants to unveil biomarkers of various prenatal conditions [31], most of the AF proteome was represented by structural proteins, followed by those necessary for fetal development, transport, blood, hormone signaling, and immune response, among others (Appendix A). Of the identified proteins, 117 were unique, whereas 306 were already reported in other studies utilizing AF (Appendix A), including the 12 most abundant proteins matching the top 15 in another study [32] (Appendix A). A heat map showing the tissue of origin for the top 10% most abundant proteins indicated that the main cluster was represented by the respiratory, gastrointestinal (GI), and urinary tracts, followed by male organs and fetal/placental proteins (Figure 1A, Appendix A). Amniocyte- and mesenchymal-stem-cell-specific proteins [31], along with proteins from the brain and cerebral cortex, were also identified, consistent with the presence of an immature blood–brain barrier (Figure 1A, Appendix A).

Among the identified metabolites, the most abundant were amino acids and their derivatives, glucose, TCA intermediaries, and lipids (Appendix A), some of which were previously identified in AF from second trimester normal pregnancies [33]. The main tissues of origin for these components were the placenta, liver, fetal brain cortex, nervous tissue, pancreas, and intestine (Appendix A), supporting the notion that AF is derived mainly from fetal tissues during this period [34].

To validate the hypothesis that any metabolic changes detected through the proteomics and metabolomics of AF may have originated mainly from the fetus vs. the mother’s metabolism, two approaches were followed. One of these approaches, by identifying proteins of fetal origin (fetus only) and X-linked proteins (fetus and mother) in the AF proteome [31], allowed us to obtain an enrichment of fetal proteins over maternal ones of ~8. The average of fetal-only proteins normalized to X-linked proteins was 116.3, while that for the fetal biomarker AFP [31,36,37] was 147.9—all significantly higher than the 14.9 for the normalized maternal proteins (Figure 1B). In parallel, the average ratio of hemoglobin gamma (HBG1 and HBG2, both of fetal origin) to that of hemoglobin beta (adult origin) was 28.4, indicating that the AF was enriched in fetal Hb, with the detected adult Hb in AF being an unavoidable consequence of the amniocentesis procedure. Taken together, the AF was enriched by 8–28 times in fetal vs. maternal proteins.

### 2.3. Impact of a Deficient Glycolysis-Derived Serine Biosynthetic Pathway in the Premutation

To test the separation of the diagnostic groups and predict outcomes based on the omics data, we used two approaches (supervised and unsupervised learning models) within the artificial intelligence and machine learning fields. For the first approach (supervised), classification problems (in our case, separating the two diagnostic groups) use an algorithm to accurately assign test (labeled) data into specific categories (i.e., premutation and non-carriers). In our case, we used the classification algorithm named partial least squares-discriminant analysis (PLS-DA; Figure 2A,C). For the second approach (unsupervised learning), the learning model uses machine learning algorithms to analyze and cluster unlabeled data sets. These algorithms discover hidden patterns in data without the need for human intervention (hence, they are “unsupervised”). We used unsupervised learning models for agglomerative hierarchical clustering (visualized here as heat maps; Figure 2B,D; Appendix A). During agglomerative clustering, data points were initially isolated as separate groupings, and then they were merged iteratively based on similarity until a minimum number of clusters was achieved. To measure similarity, we used Ward’s linkage. This method states that the distance between two clusters is defined by the increase in the combined error sum of squares after the clusters are merged. This distance was calculated using the Euclidean algorithm, as this is the most common metric used to calculate these parameters. Surprisingly, the inspection of the plots obtained with supervised and unsupervised learning models showed a marked distinction between carriers and non-carriers, as not all pediatric carriers exhibit early and/or overt metabolic and clinical phenotypes [6,7].

Pathway analysis of a joint AF metabolome–proteome revealed upregulation of aminoacyl-tRNA biosynthesis, as well as the metabolism of the following amino acids: sulfur-containing (Cys and Met), branched-chain (Val, Ile, Leu), aromatic (Phe, Tyr and Trp), and Arg and Pro. The downregulated pathways included, among others, energy supply (glycolysis and TCA cycle); metabolism of Ala, Gly, Ser, Thr, Asp, and Glu; the pentose phosphate pathway; glutathione metabolism; and pantothenate and coenzyme A biosynthesis (Figure 3).

Conditions of increased oxidative stress were incurred by the downregulation of the pentose phosphate pathway (Figure 3B), glutathione metabolism (GPX3, GSTP1; Figure 3B, Appendix A), antioxidant defenses (SOD3, HP, TXN; Appendix A), and lower contents of Glu and selenocysteine (glutathione; Appendix A), likely offsetting the antioxidant defense status.

Impaired glycolysis was gathered by the lower expression of GAPDH and the downstream enzymes of the glycolytic pathway (PGK1, PGAM1, and ENO1; Figure 4A). These deficits resulted in lower production of both pyruvate and lactate, the latter further compounded by the lower expression of LDHA/B (Figure 4A).

To avoid algorithm and pathway database bias, we repeated the joint analysis utilizing different algorithms (utilizing as input only those metabolites and proteins with VIP scores of 0.8 or above, and the WikiPathways and REACTOME pathway databases). This analysis resulted in 34 modules (Appendix A), several overlapping with those already identified under Figure 3, and most overlapping (albeit to a lesser extent) with those identified in bio-samples from adult PM carriers [10,23,41,42,43].

It could be argued that the association between maternal age and dizygosity incidence—leading to age-dependent epigenetics (e.g., X chromosome inactivation or imprinting) or the association between the genetics of dizygotic twinning and cerebral asymmetry—may have influenced the results presented here. However, support for the non-concordance of the biological outcomes between dizygotic twins was performed by principal components analysis (Appendix A). For this reason, data obtained on AF from the two sets of non-identical twins were included (note: analysis of the data excluding the two sets of non-identical twins or including one of each set did not modify the conclusions of the study).

Taken together, these results point to a general decline in cellular energy metabolism in carriers, encompassing both glycolysis and oxidative phosphorylation linked to deficiencies in the Ser and Gly biosynthetic pathways. This is supported by the major decline in the levels of Asp, Glu, and Ser (Appendix A)—amino acids that are biochemically linked to AKG by transamination (Glu/AKG or AKG-Asp/oxaloacetate) or via Ser biosynthesis (AKG/Glu-Ser). In agreement with this concept, Asp has been shown to play a role in the regulation of Ser uptake and metabolism, as well as downstream pathways, in rapidly proliferating cells [44].

Notably, PGK1 coordinates glycolysis with the TCA cycle (via mitochondrial translocation) [45], autophagy [46], and endoplasmic reticulum (ER) stress response [47], including the production of 3-phosphoglycerate, the substrate for the ultimate biosynthesis of Ser and Gly. Based on the anaplerotic role that the Ser biosynthetic pathway has on the Krebs cycle [48], the downregulation of Ser synthesis suggests a limited AKG supply, which would limit mitochondrial energy production (Figure 4A). In this regard, the direct correlation of AKG with TCA intermediates downstream from AKG dehydrogenase (i.e., malate, fumarate, succinate), as well as Glu, Ser, and Gly levels, and the inverse correlation with Gln and pantothenate (Appendix A) reinforces the biochemical link between AKG and both processes—the TCA cycle and the Ser biosynthetic pathway. Aside from providing AKG, the Ser-dependent pathways also provide intermediates for the synthesis of creatine (requiring Gly) and carnitine (Appendix A), further impacting the cellular energy management.

It could be claimed that the impact of a non-essential amino acid such as Ser on cellular metabolism is negligible; however, its role is critical during CNS neurodevelopment, based on its low blood–brain barrier permeability [49], and the high demand of prenatal energy-dependent synthesis of nucleotide precursors [50] and myelin, among which are key brain sphingolipids and gangliosides essential for dendritic outgrowth [51]. More importantly, Ser is involved in neurotransmission through its involvement in the GABA shunt (AKG, Glu, succinate; Appendix A) and synthesis of glycine and D-Ser (direct co-agonists of the NMDA receptor along with Glu), thereby fine-tuning synapses, neuronal plasticity, and excitotoxicity [52]. It is critical to highlight that sustaining biosynthetic processes while minimizing ROS-mediated damage (especially when derived from mitochondria) is key to neurodevelopment [53]. As such, the combination of decreased antioxidant defenses with increased mitochondrial ROS resulting from mitochondrial dysfunction in carriers may impact cellular proteostasis. In this regard, evidence for a proteotoxic insult was supported by the higher levels of aconitate, likely the result of ROS-mediated aconitase inactivation [54,55] (Figure 4A).

Further independent support for these findings was provided by the significant overlap of dysregulated pathways between the metabolic footprints observed in AF from fetal carriers and those produced by stressors relevant to energy management (hypoxia, oxygen, and glucose deprivation or OGD—mitochondrial stressors) and proteotoxicity (endoplasmic reticulum stress; Appendix A).

The prenatal PM metabolic footprint was aligned mainly with conditions related to energy stress driven by HIF-1α-independent signaling (early OGD response and PHD3 silencing), mitochondrial uncouplers and inhibitors, and ER stress (Appendix A). A major contribution to the HIF-1α-dependent activation in the metabolic changes from carriers was precluded based on the lower levels of 2-hydroxyglutarate (associated with both lower PHGDH [56] and LDH [57]). Moreover, the prenatal PM metabolic footprint was analogous not only to complex diseases associated with underlying MD (schizophrenia and Alzheimer’s disease; Appendix A), but also with others directly and indirectly related to the Ser biosynthetic pathway (e.g., deficiencies in lipoyltransferase, AKG dehydrogenase, PSAT1, PSP and PGHDH; Appendix A, in bold).

### 2.4. Independent Validation of the Prenatal Metabolic Footprint of the PM

To prevent bias-corrected estimates of model performance [58], the prenatal metabolic footprint was independently validated by utilizing one internal (AF) and two external cohorts. The two external cohorts from PM carriers and non-carriers—all obtained at a center located in California (Appendix A)—included proteomics from 25 primary dermal fibroblasts and metabolomics from 39 plasma samples (characteristics of donors under Appendix A).

A receiver operating characteristic curve for the predictive performance of the prenatal PM metabolic footprint was built utilizing two-thirds of the AF samples (randomly selected) and the top variable in importance (VIP) predictors from both proteomic and metabolomic data (Appendix A), and then further tested by incorporating the holdout samples. The performance of the model was significant (100 cross-validations and *p* < 0.01), with an accuracy of 77.78%.

When the model was built with either the proteomics data of all AF samples, with proteomics from fibroblasts used as holdout samples (Appendix A), or the metabolomics from all AF samples and the metabolomics of plasma samples (Appendix A), the performance of both models was significant after 100 cross-validations (*p* < 0.01 and < 0.05 for proteomics and metabolomics; Appendix A, respectively), with an accuracy of the models of 77.8% and 65.45% for the proteomics- and metabolomics-based data, respectively, despite the differences in sex (all AF being from males), developmental stage, age (prenatal vs. adults), and tissues (AF vs. fibroblasts and plasma). Taken together, these results indicate that the metabolic footprint of AF from carriers, albeit with different degrees of severity, matches those of other biological matrices obtained from adult carriers.

### 2.5. Improving Ser Status Rescues Mitochondrial Function in the PM

Based on our findings, we hypothesized that RNA- and protein-mediated toxicity, as a result of the *FMR1* CGG expansion, impacted the Ser pathway. We reasoned that if glycolysis-derived Ser was critical in providing AKG for the TCA cycle in the PM, then mitochondrial outcomes of non-carriers with a halted Ser biosynthesis should result in mitochondrial outcomes like those of the PM. Conversely, if Ser biosynthesis were the rate-limiting step for providing AKG, then mitochondrial outcomes in the PM should respond favorably to the supplementation of AKG precursors. As a proof of concept, we tested these options by utilizing umbilical cord fibroblasts (UCFs)—tissues reflecting prenatal periods as close as possible to those of the AF—from non-carriers and carriers (two for each). Compared to non-carriers, UCFs from carriers presented mitochondrial hypofunction. This was characterized by a lower fraction of the maximum mitochondrial capacity to synthesize ATP (IRC or index of respiratory oxygen uptake coupled with ATP production [59]), lower maximum electron transport capacity (or State 3u evaluated under uncoupling conditions), and lower capacity to synthesize ATP (State 3 evaluated under physiological conditions; Figure 4B black bars). These results confirmed the presence of MD in the PM even in samples obtained at these early stages of life.

Inhibition of the endogenous synthesis of Ser in non-carrier UCFs with WQ-2101 (inhibitor of PHGDH [60]) resulted in mitochondrial deficits similar to those obtained from carriers (Figure 4B open bars), regardless of the presence of Ser and Gly in the cell growth medium, highlighting the relevance of this anaplerotic pathway during early development periods. Supplementation of carrier UCFs grown in media with glucose (and Ser and Gly) and AKG precursors (Gln, AKG, or their combination), compared to non-carrier UCFs grown in glucose only, resulted in a significant improvement of coupling between electron transport and ATP production (RCR; AKG, and combination), IRC (Gln, AKG, and combination), State 3u (AKG and combination), proton leak, and ROS production (State 4o; AKG and combination), and State 3 (AKG) (Figure 4C). Addition of glucose, Gln, and AKG improved the RCR (respiratory control ratio), IRC, State 3, and State 4o compared to glucose alone (Figure 4C).

Further confirmation on the contribution of this metabolic pathway in UCFs was obtained through metabolomics analysis (Table 2). UCFs from non-carriers and carriers grown in glucose and supplemented with Gln showed increases in the [ATP]/[AMP] ratio, with decreased glycolytic and TCA cycle fluxes (Table 2).

When the media were supplemented with the cell-permeant AKG, cells from carriers and from non-carriers with or without the PHGDH inhibitor responded similarly, showing increased ATP:AMP ratios and moderate impact on the glycolytic flux, but with significant improvements in the TCA cycle. Cells from non-carriers treated with the PGDH inhibitor resulted in a significant buildup of glucose 6-phosphate and 3-phosphoglycerate, thereby confirming a block in the branch that leads to the formation of AKG, Ser, and Gly from 3-phosphoglycerate. Similar metabolomics results were obtained when both Gln and AKG were used to supplement the media with glucose, indicating that the greatest effect was achieved by the cell-permeant AKG, as shown by the mitochondrial functional outcomes (Figure 4C).

These results are consistent with those obtained with AF in which the disease gene pathway analysis run with the joint input of metabolites and proteins indicated deficiencies in the AKG dehydrogenase and Ser biosynthetic pathways (bolded in Appendix A), and AKG as the most discriminating metabolite, along with other TCA intermediates (malate, fumarate, succinate; Appendix A). It is important to note that TCA intermediates’ concentrations are not only driven by the activity of the TCA cycle (e.g., AKG-derived succinate), but also by other anaplerotic reactions (e.g., succinate levels in carriers without Gln vs. those of other TCA intermediates; Table 2). In this regard, succinate concentrations may also be sustained by (for example) the catabolism of fatty acids with uneven numbers of carbons through the formation of propionate, as well as by GABA metabolism. In support of this premise, the correlation coefficient for AKG with succinate was lower (and less significant) than that for AKG with malate or fumarate (Appendix A). Therefore, it is important to be cognizant of the ratios among metabolites, as well as their absolute concentrations, by utilizing analyses that are comprehensive and integrated, such as those used in systems biology.

We are cognizant that metabolic changes obtained with different cell types or tissues (in our case AF, UFC, PBMCs, or skin fibroblasts) may present different degrees of severity, as is usually seen with mitochondrial disorders, further complicated by the phenotypic threshold effect [20,61,62]; however, results obtained with these diverse cells and tissues all converged at showing deficits in the anaplerosis of AKG by the glycolysis-driven Ser biosynthetic pathway. Taken together, these results clearly show that cells from carriers can respond to AKG supplementation, thereby improving the performance of the Krebs cycle and mitochondrial energy output. Furthermore, they support the involvement of the Ser pathway as a critical anaplerotic source for the Krebs cycle in PM carriers, especially during perinatal periods and, for the first time, in non-tumoral tissues.

## 3. Materials and Methods

### 3.1. Subjects’ Demographics; FMR1 Repeat Sizing and Structure

Fifty-two pregnant women, all carriers of the PM, gave consent for any AF samples remaining after prenatal fragile X diagnosis to be used for research. All available demographic and genetic data are shown in Table 1. The *FMR1* repeat sizing was performed by PCR analysis using AmplideX PCR CE/*FMR1* (Asuragen, Inc., Austin, TX, USA), and the products were separated by capillary electrophoresis with Applied Biosystems 3130. The AGG interruption patterns were determined using either AmplideX PCR CE/*FMR1* at the New York State Institute for Basic Research in Developmental Disabilities (New York, NY, USA), or Xpansion Interpreter (Asuragen, Inc., Austin, TX, USA). Changes in allele repeat length on transmission were determined by comparison of parental and transmitted alleles in parallel PCR capillary electrophoretic analyses. An unstable transmission was defined as a change of at least one repeat from parent to child [24].

### 3.2. Metabolomic and Proteomic Evaluation

Metabolomics and proteomics analyses of 54 AF samples, external validation cohorts constituted by dermal fibroblasts and plasma samples obtained from non-carriers and carriers (Appendix A), and UCFs were carried out essentially as previously described [6,7,8]. More details are reported in the Appendix A.

### 3.3. Cell Culture Conditions and Mitochondrial Outcomes

Fibroblasts (P4) obtained from the umbilical cords of 2 controls (both males, CGG repeats = 25 and 27) and 2 PM carriers (one female, CGG = 69; one male, CGG = 68) were thawed in medium consisting of a 1:1 ratio of AmnioMAX-C100 and RPMI-1640 basal medium supplemented with 10% fetal bovine serum (FBS, HyClone), 1× penicillin/streptomycin (Gibco), and 1% non-essential amino acids (Gibco), as previously described [10]. At 80–90% confluency, cells were switched to RPMI-1640 medium supplemented with 15% FBS in order to prevent putative AmnioMAX-associated metabolic changes. When control fibroblasts reached ~50% confluence, a subset was treated for 24 h with 25 µM WQ-2101 or vehicle (DMSO, 0.0026%) in the same medium. At 75–80% confluency, all cells (controls and PM) were detached by trypsinization, pelleted at 300× *g*, washed twice in RPMI media without FBS, and resuspended in RPMI supplemented respectively with (1) 4 mM Glucose, (2) 2 mM Glutamine, (3) 3 mM DMAKG, and (4) a combination of the four. Viability was evaluated by trypan blue assay with a TC20 counter (Bio-Rad). For all cell lines, viability was 97–99%.

Oxygen consumption was immediately evaluated in intact cells using a Clark-type oxygen electrode (Hansatech, King’s Lynn, UK), as previously described [10]. ATP-linked oxygen uptake (or State-3-dependent oxygen uptake) was calculated as the difference between basal and oligomycin-induced State 4 oxygen uptake rates; State 4o is the residual respiration after inhibition of ATP synthesis with the ATPase-specific inhibitor oligomycin (2 μM); maximal respiratory capacity, or State 3u, is described as the oxygen uptake rate in the presence of the uncoupler carbonyl cyanide-4-(trifluoromethoxy)phenylhydrazone (FCCP) (40 nM); respiratory control ratio (RCR) was calculated as the ratio between States 3 and 4o; index of respiratory capacity (IRC) was calculated as the difference between State 3 and State 4o normalized by that of State 3u.

### 3.4. Statistical Analysis

Spectral counting was utilized to quantify the proteins from mass spectrometry analysis. Both protein and metabolite levels were normalized to pooled samples from pregnancies of non-carrier fetuses; the features were normalized by utilizing the quantile and then auto scaled (mean-centered and divided by the standard deviation of each variable). Combined pathway analysis was performed using MetaboAnalyst [38]. The threshold for statistical significance was set at the 5% level, unless otherwise indicated. For mitochondrial outcomes, statistics was performed using Student’s t test between two groups, and ANOVA followed by Tukey’s post-hoc test when comparing multiple groups. Other details are indicated in the figure legends or the Appendix A.

## 4. Conclusions

A critical deficiency in the Ser biosynthetic pathway was noted by a thorough joint analysis of the proteome and metabolome of AF from PM-carrier fetuses. This deficiency not only impacted energy management (supply of AKG to mitochondria, creatine synthesis), but also has the potential to affect other biosynthetic pathways relevant to neurodevelopment (lipid and protein syntheses, detoxification). It could be argued that mild prenatal changes in mitochondrial function would not result in fetal metabolic changes in utero, as this environment is considered “hypoxic”. However, a wealth of studies reported embryonic lethality at equivalent gestational ages for homozygous knockouts of respiratory chain subunits (NDUFA5 [63], SDHD [64], and RISP-Rieske iron sulfur protein [65]), ancillary factors (TMEM70 [66], NDUFS4 [67], COX15 [68], COX17, and SCO2 [69,70]), and mitochondrial antioxidant defenses (TRX-2 [71,72]), highlighting the critical role of mitochondria during pregnancy. Furthermore, the fact that aerobic production of ATP increases significantly from the second through the third trimesters in preparation for the higher atmospheric oxygen does not imply that during this period mitochondria do not play any role in fetal development. The amniotic *p*O_2_ between weeks 15 and 17 is <10 mmHg (10–15 μM oxygen [73]), able to sustain oxidative phosphorylation throughout development, as this oxygen tension is well above the *K_m_* for oxygen of cytochrome *c* oxidase (*K_m_* = 0.9 M [74]). However, it could be inferred that any (even mild) mitochondrial changes in the offspring would be more evident soon after birth. This possibility is consistent with the current view that mitochondrial dysfunction begins long before pathological signs are evident. As such, it is possible that some of these prenatal mitochondrial changes may be magnified after birth and persist into adulthood.

While decreasing energy provision by downregulating the Ser biosynthetic pathway limits macromolecule synthesis to reduce the load of ROS-mediated damage to misfolding-prone proteins, this mechanism also puts a brake on CNS cell growth and differentiation at a critical stage of brain development. The encouraging results of Ser supplementation to treat individuals with PHGDH deficiency [75], ALS [76], and Alzheimer’s disease [77], along with our findings on downregulation of the Ser biosynthetic pathway and the recovery of mitochondrial function by using AKG precursors, raise the possibility of introducing precursors of AKG, including Ser supplementation, as a ready-to-use intervention during pregnancy to minimize the risk of children developing atypical neurobehavioral and emotional phenotypes or, later in life, the occurrence of the neurodegenerative disease FXTAS.

## Figures and Tables

**Figure 1 ijms-22-05886-f001:**
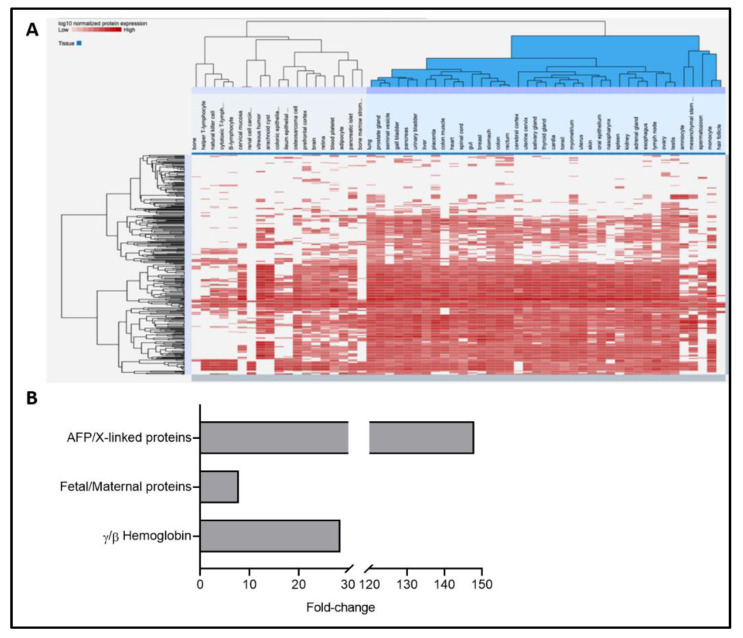
Confirmation of the fetal origin of AF. A heat map showing tissue representation in AF was obtained with the top 10% most abundant proteins from the AF proteomes. Analysis was performed as described in [35]. (**A**) Respiratory (lungs, oral epithelium), GI (esophagus, salivary gland, colon, rectum, gut, pancreas, stomach, liver, gall bladder), and urinary tract (kidney, urinary bladder) tissues were identified, along with male organs (e.g., spermatozoon, testis, seminal vesicles, prostate gland), as well as fetal and placental proteins. Amniocyte- and mesenchymal-stem-cell-specific proteins, and proteins from the brain and cerebral cortex, were also identified. (**B**) Those proteins predominantly or exclusively fetal were AFP, SI, LCT, CST3, SERPINA1, CP, TF, and ORM1; those X-linked were ARMCX4, BGN, CD99L2, CHRDL1, EFNB1, FLNA, IGSF1, MSN, MXRA5, PCSK1N, PLS3, SERPINA7, SRPX, TIMP1, TMSB4X, and VSIG41; and those exclusively of maternal origin were HP and HPX [31]. The ratios of fetal markers to X-linked markers (mother and male fetus), and of alpha-fetoprotein (AFP) normalized to X-linked proteins, were 7.8- and 147.9-fold, respectively, confirming the overwhelmingly higher fetal contribution in AF samples. The ratio of fetal hemoglobin (γ-Hb) to that of adult hemoglobin (β-Hb) was 28.4-fold in AF samples.

**Figure 2 ijms-22-05886-f002:**
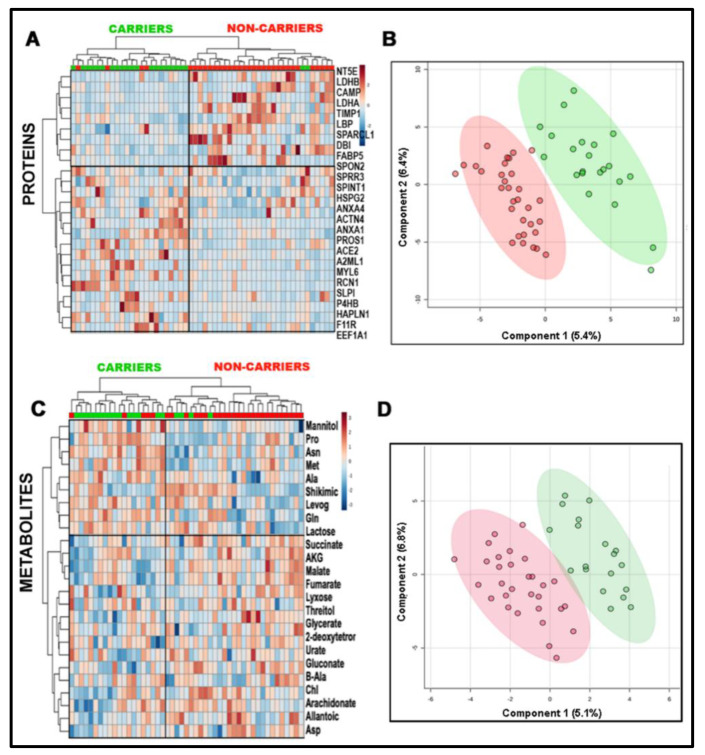
Classification of subjects by utilizing omics data, based on supervised and unsupervised learning models. An unsupervised agglomerative hierarchical clustering was performed using a Euclidean distance measure and Ward’s clustering algorithm; this analysis was visualized by showing the top 25 features in a heat map according to Student’s *t* test for proteins and metabolites between non-carriers and PM carriers (**A**,**C**). A supervised analysis (PLS-DA) was performed to identify the proteins and metabolites that separated the most carriers (green) from non-carriers (red) (**B**,**D**; see details in Appendix A).

**Figure 3 ijms-22-05886-f003:**
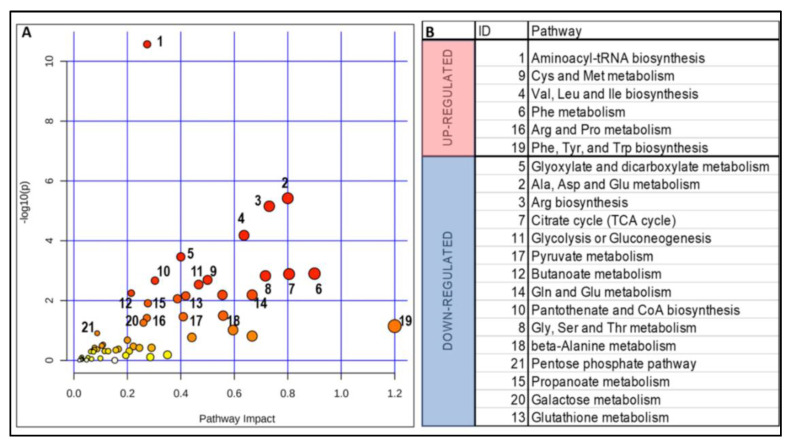
Omics-based pathway analysis of amniotic fluid. (**A**) Analysis was performed with the joint pathway module from MetaboAnalyst [38], which allows for an integrated approach to results obtained from combined metabolomics and protein expression studies conducted under the same experimental conditions. Enrichment analysis was based on the hypergeometric features and utilizing the topology based on the degree centrality (which evaluates the number of links that connect anode) within a pathway. The database utilized was KEGG, and the integration method combined the queries. Results were plotted as a function of impact within the pathway and significance. Labeled pathways have an FDR < 0.05. (**B**) Differentially expressed pathways in AF supernatants from carriers and non-carrier fetuses. These results were obtained by using the metabolomics and proteomics data (filtered by log2 FC >1 or <−1 and *p* < 0.05) as input data, and analyzing them using the pathway modeling software PathVisio version 3.0 [39], which computes Z-scores as well as permutated *p*-values. This analysis was repeated utilizing proteomics data alone and employing the STRING database version 11, which specifically asks for proteome scale input, with each protein having an associated numerical value (log2 FC) [40]. Of the available methods for searching functional enrichments in such a set, we chose a permutation-based, non-parametric test that computes, for each protein set to be tested, the average of all values provided by our dataset for the constituent proteins. This average was then compared against averages of randomized gene sets of the same size. Multiple testing correction was applied separately within each functional classification framework (KEGG), in accordance with the method of Benjamini and Hochberg, but not across these frameworks, as there is significant overlap between them.

**Figure 4 ijms-22-05886-f004:**
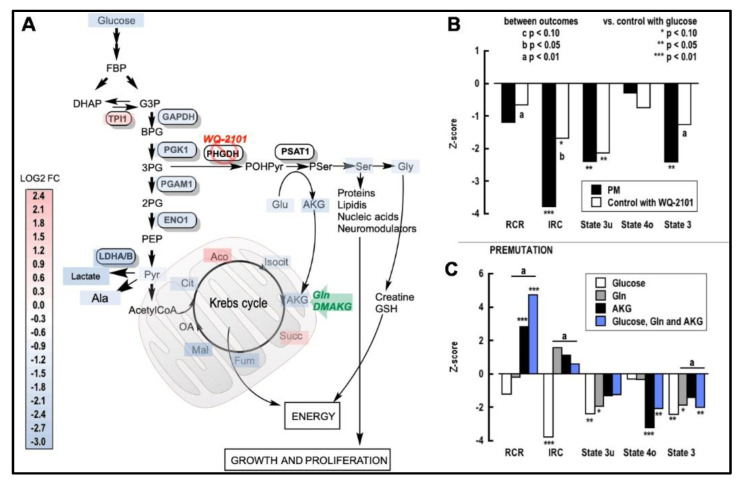
Role of Ser in the PM energy metabolism. (**A**) Metabolites and proteins identified in this study associated with glycolysis and the TCA cycle, whose levels were different between diagnostic groups, are shown in red (upregulated) or blue (downregulated). Intervention with dimethyl α-ketoglutarate (DMAKG), a membrane-permeable ester of AKG, is shown with a green arrow. Inhibition of the Ser biosynthetic pathway by WQ-2101, a potent and selective competing allosteric inhibitor of phosphoglycerate dehydrogenase (PHGDH), is shown in red. (**B**) Treatment with 25 µM WQ-2101 was carried out on 2 control umbilical cord fibroblasts, as described in the Methods section. After 24 h, cells were evaluated for oxygen consumption in the presence of 4 mM glucose (State 3), followed by addition of oligomycin (State 4o) and FCCP (State 3u). The same outcomes were recorded in 2 PM umbilical cords under the same conditions. Statistical analysis was performed using Student’s *t* test, for the comparison between PM and WQ-2101-treated control fibroblasts. Multiple groups were analyzed using ANOVA followed by Tukey’s post-hoc test. (**C**) The same mitochondrial outcomes were tested in PM umbilical cord fibroblasts in the presence of glucose (4 mM), Gln (2 mM), DMAKG (3 mM), and the combination of the three substrates. Bar graphs are shown as Z-scores. RCR: respiratory control ratio; IRC: index of respiratory capacity.

**Table 1 ijms-22-05886-t001:** Characteristics of the cohort and samples available for this study.

Non-Carrier Fetuses					Carrier Fetuses				
Maternal Age ^a^ (y)	Gestational Age (Weeks)	Maternal CGG Repeat Size	Fetal CGG Repeat Size	Fetal Repeat Structure ^b^	Maternal Age ^a^ (y)	Gestational Age (Weeks)	Maternal CGG Repeat Size	Fetal CGG Repeat Size	Fetal Repeat Structure ^b^
25.5	17.3	29, 92	29	29 ^c^	17.8	16.4	37, 77	88	9A78
25.5	17.3	29, 92	29	29 ^c^	18.0	16.4	30, 62	62	9A9A42
25.8	16.7	29, 53	29	9A9A9	20.4	16.4	30, 55	55	9A45
27.4	18.2	29, 55	29	9A9A9	20.6	16.3	29, 60	66	66
28.5	18.3	30, 63	30	10A9A9	23.6	21.2	30, 56	57	9A9A37
30.0	15.7	30, 132	30	10A9A9	24.4	16.5	23, 57	58	10A47
30.5	17.5	26, 59	26	26	26.2	17.1	31, 58	63	63
30.7	16.3	30, 68	30	10A9A9	27.0	15.7	29, 60	61	9A51
31.2	16.9	29, 200	29	9A9A9	28.4	17.2	30, 60	63	9A53
31.3	15.9	29, 89	29	9A9A9	29.3	16.7	23, 63	68	9A58
31.5	16.0	29, 68	29	9A19	30.0	16.0	31, 57	58	10A47
31.6	20.3	40, 65	40	10A29	31.3	16.5	33, 57	57	9A9A37
31.8	16.6	20, 56	20	10A9	31.5	16.4	23, 67	78	9A68
31.9	15.5	30, 58	30	10A9A9	32.8	16.0	31, 74	63	9A53
32.0	17.1	29, 56	29	9A9A9	33.0	16.3	23, 59	67	67
32.8	16.1	29, 53	29	9A9A9	33.3	20.0	30, 57	59	10A48
33.2	18.2	29, 71	29	9A9A9	33.4	13.3	30, 82	141,157 ^e^	9A9A121
33.8	16.7	21, 58	21	10A10	33.8	17.0	30, 63	63	9A9A43
34.3	17.1	30, 73	30	10A9A9	35.4	16.0	20, 69	72	9A9A52
34.7	16.3	32, 60	32	9A12A9	35.7	16.5	31, 61	66	9A56
35.1	21.0	30, 59	30	10A9A9	40.7	15.1	30, 72	84	84
36.1	18.9	29, 68	29	9A19	42.1	17.7	30, 58	58	9A9A38
37.1	15.7	20, 105	20	10A9 ^d^					
37.1	15.7	20, 105	20	10A9 ^d^					
37.9	15.3	30, 63	30	10A9A9					
37.9	17.4	20, 62	20	10A9					
38.9	16.1	30, 75	30	10A9A9					
39.1	16.4	29, 59	29	9A9A9					
39.2	15.9	30, 58	30	10A9A9					
40.1	18.0	29, 56	29	9A9A9					
40.7	16.0	31, 44	44	9A9A24					

^a^: Maternal age at collection; ^b^: numbers represent the number of CGG triplets, and A represents an AGG interruption. ^c,d^: identical letters indicate twins with > 80% likelihood of dizygosity (see Discussion for more details); ^e^: smear was also noted in this sample.

**Table 2 ijms-22-05886-t002:** Effect of Gln and AKG on the metabolomics of umbilical cord fibroblasts.

	Diagnosis	Non-Carrier	Carrier	Non-Carrier	Non-Carrier	Carrier	Non-Carrier
	Substrate	Gln			AKG		
	Inhibitor of PHGDH	no	no	yes	no	no	yes
**Energy Status**	[ATP]/[AMP] ratio	2.31	1.88	2.15	1.18	1.48	1.23
	AMP	−2.73	−1.53	−2.30	−1.61	−1.10	1.04
**Glycolysis**	Glucose	−2.86	−2.88	−2.40	−1.42	1.24	−1.36
	G6P	−3.09	−3.30	−1.87	1.15	1.17	1.79
	3PG	−2.19	−2.71	−2.15	1.02	1.13	1.70
	Lactate	−2.51	−2.50	−1.66	−1.62	−2.29	−1.40
	Ala	−2.61	−1.48	−3.04	−1.75	−1.35	−1.46
**TCA Cycle**	Citrate	−2.42	−2.21	−1.84	1.33	2.77	−1.17
	AKG	−3.16	−3.20	−3.04	136.28	51.43	235.19
	Succinate	−1.80	4.68	−1.76	5.90	12.20	5.26
	Fumarate	−2.06	−1.67	−2.57	3.74	2.15	3.54
	Malate	1.06	−1.57	−1.40	12.65	2.41	10.00
**Anaplerotic Substrates**	AKG	−3.16	−3.20	−3.04	136.28	51.43	235.19
	Asp	−2.65	−1.78	−2.91	−6.79	−5.99	−3.61
	Glu	−2.44	−1.57	−1.95	−1.11	−1.09	1.29

Metabolomics was performed on UCFs from age- and sex-matched controls and carriers. Values represent the log2 FC normalized to glucose alone. Significant changes are indicated in red (up) or blue (down), with *p* < 0.05. WQ-2101: PHGDH inhibitor.

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
