# Peer review of "Deficits in Prenatal Serine Biosynthesis Underlie the Mitochondrial Dysfunction Associated with the Autism-Linked FMR1 Gene"

_ijms, 2021, doi:10.3390/ijms22115886_

Round 1
Reviewer 1 Report
This is a very interesting manuscript with higher level statistic and proteomics where clustering and conclusions may be better judged by a clinician in the proteomics field. The conclusions are surprising but support evidence of in utero mitochondrail dysfuction in PM carriers and increased glycolysis as has been reported in tissue and brain samples for adult premutation carriers and FXTAS patients. It is interesting at a proteinomic level this has parallels with Alzheimers and Schizophrenia. The level of impact this dysfunction has on the growing brain for what is typically thought to be late adult phenotype remains to be seen and how this would be influenced by antenatal administration of Serine is an interesting hypothesis. It is possible this may influence neuropsychiatric and learning phenotypes in PM carriers and may be a very important observation. The authors in their experiements never treated cord fibroblasts with Serine - are their technical reason's not to attempt this or will the influence be seen over a longer timeframe to test this hypothesis.
My comments are
Introduction PM often expand in one generation to full mutations - this is relatively rare for the majority of PM carriers who have low repeat sizes - suggest "may expand"
PM carriers do not show show higher levels of severe developmental delay than their peer group (Kraan et al Genetics in Medicine 2018: 20, 1627–1634). There is some observer bias in the publications referenced - this should be clarified.
Layout of table 1. The adjacent carriers non carriers in the table are these cases matched and on what basis - why is the table laid out in that way.
You indicate most PM's change in size when transmitted by +1 - would suggest quote >+1 as thats uusally consider error of measurement. This would mean less than 50% expand when transmitted.
Major concern Figure 2C - the number of green samples on horizontal access labeled carriers looks greater than non carriers which is not in keeping with sample sizes described or figure 2A which has fewer carriers. Please explain why this figure has more carriers than non carriers and if the data is inverted it reverses your conclusions about the metabolic pathway so I am now quite confused. This needs to be explained or rectified.
Figure 3 - how do you read what is upregulated versus down the Logfold change of metabolite versus pathway impact is not clear - perhaps help reader - some are upregulated having a high impact but low fold change (all positive) and others have a high fold change low impact. Not clear to the non proteomic reader. Suggest clarify if possible.
The last question did you observe any variation in impact based on size of CGG expansion. Other studies have suggested more marked changes with greater CGG size - can you put data into two bins for CGG expansion size /low and high and see if theres a relative trend with greater size showing worsening mitochondrial insult
Author Response
Q1. This is a very interesting manuscript with higher level statistic and proteomics where clustering and conclusions may be better judged by a clinician in the proteomics field. The conclusions are surprising but support evidence of in utero mitochondrail dysfuction in PM carriers and increased glycolysis as has been reported in tissue and brain samples for adult premutation carriers and FXTAS patients. It is interesting at a proteinomic level this has parallels with Alzheimers and Schizophrenia. The level of impact this dysfunction has on the growing brain for what is typically thought to be late adult phenotype remains to be seen and how this would be influenced by antenatal administration of Serine is an interesting hypothesis. It is possible this may influence neuropsychiatric and learning phenotypes in PM carriers and may be a very important observation. The authors in their experiements never treated cord fibroblasts with Serine - are their technical reason's not to attempt this or will the influence be seen over a longer timeframe to test this hypothesis.
A1. We thank the reviewer for the positive feedback and very helpful suggestions. Please see our responses to individual queries below.
In regards to the question why we did not try to supplement umbilical cord fibroblasts with Serine is based on the fact that this amino acid’s concentration may or may not be deficient prenatally. But the main issue is that the pathway that leads to Serine acts as an anaplerotic reaction to the Krebs’ cycle by generating alpha-ketoglutarate. Thus, providing the cell-permeant alpha-ketoglutarate analog was the best experimental approach to rescue mitochondrial dysfunction by supplying this intermediate to the TCA cycle.
Q2. Introduction PM often expand in one generation to full mutations - this is relatively rare for the majority of PM carriers who have low repeat sizes - suggest "may expand"
A2. Thank you for this comment. We have changed the sentence as suggested.
Q3. PM carriers do not show higher levels of severe developmental delay than their peer group (Kraan et al Genetics in Medicine volume 2018: 20, pages1627–1634). There is some observer bias in the publications referenced - this should be clarified.
A3. Although a thorough discussion of the findings relative to developmental disabilities in PM children is outside of the scope of this study, we have added a short paragraph in the Introduction to address the issue raised by the reviewer.
Q4. Layout of table 1. The adjacent carriers non carriers in the table are these cases matched and on what basis - why is the table laid out in that way.
A4. We apologize for the formatting oversight (when performing copy and paste somehow the Table rows were misplaced). Table 1 is now correctly formatted.
Q5. You indicate most PM's change in size when transmitted by +1 - would suggest quote >+1 as that’s usually consider error of measurement. This would mean less than 50% expand when transmitted.
A5. An important component of this study was the accuracy of fragile X clinical diagnostic molecular testing that does not require verification by additional testing. Dr. Nolin’s group has performed >5,000 fragile X prenatal studies. The rigorous quality control standards for diagnostic testing include analyzing fetal, maternal, and paternal samples in parallel to detect very low levels of maternal cell contamination and confirmation of prenatal analyses after birth. Of those who provided postnatal blood samples (~10%), all results confirmed the prenatal diagnoses. Dr. Nolin used some of these very same AF to determine AGG interspersion patterns to investigate their effect on risk for expansion [1-3]. While the exact mechanism is still unknown, these AGG interruptions do not appear to play a direct role in the transcription and/or translation of the FMR1 gene [4,5], they increase significantly FMR1 genetic stability and both the CGG repeat length and AGG interruptions do lower the risk of expansion to a full mutation during maternal transmission of a PM allele [3,6,7]. As Dr. Nolin is a pioneer and expert in the role of AGG interruptions in the instability of the premutation transmission, we utilized here her and the world-wide accepted definition of instability as the change of at least one repeat (one CGG repeat) from parent to child[8].
Q6. Major concern Figure 2C - the number of green samples on horizontal access labeled carriers looks greater than non carriers which is not in keeping with sample sizes described or figure 2A which has fewer carriers. Please explain why this figure has more carriers than non carriers and if the data is inverted it reverses your conclusions about the metabolic pathway so I am now quite confused. This needs to be explained or rectified.
A6. One analysis was run with the labels “PM” and “TD” for which the software used default coloring (red for PM and green for TD by following alphabetical order). In another analyses, we labeled the samples as having diagnosis 1 (non-carriers) and diagnosis 2 (carriers). The software’s default coloring used red on those with diagnosis 1 and green in those with diagnosis 2. This resulted in the color swap observed by the keen reviewer on the Figure 2. We apologize for any confusion that this may have caused, and we thank you for picking this up. Please see revised Figure 2.
Q7. Figure 3 - how do you read what is upregulated versus down the Logfold change of metabolite versus pathway impact is not clear - perhaps help reader - some are upregulated having a high impact but low fold change (all positive) and others have a high fold change low impact. Not clear to the non proteomic reader. Suggest clarify if possible.
A7. We have used two approaches for establishing the up- and down-regulation of biological pathways. For the first approach and by using proteomics data only, we used STRING database version 11 which specifically asks for genome- or proteome scale input, with each protein or gene having an associated numerical value (Log2 fold change) [9]. Of the available methods for searching functional enrichments in such a set, we chose a permutation-based, non-parametric test that computes for each gene set to be tested, the average of all values provided by the our dataset for the constituent proteins. This average was then compared against averages of randomized gene sets of the same size. Multiple testing correction was applied separately within each functional classification framework (KEGG), according to Benjamini and Hochberg, but not across these frameworks as there is significant overlap between them. The second approach and by using as input both the metabolomics and proteomics data (filtered by log2FC >1 or <-1 and p <0.05), we used the pathway modeling software Pathvisio version 3.0 [10] which computes Z scores as well as permutated p-values.
We have added a summary of this information under Section 2.3. and Figure 3 legend.
Q8. The last question did you observe any variation in impact based on size of CGG expansion. Other studies have suggested more marked changes with greater CGG size - can you put data into two bins for CGG expansion size /low and high and see if theres a relative trend with greater size showing worsening mitochondrial insult
A8. As reported also in the text of the originally submitted manuscript (page 3), since a direct correlation had been observed by several studies between CGG repeat length and increased oxidative stress as well as mitochondrial dysfunction, for the current study we selected mothers with similar CGG expansions to eliminate this putative “confounding” variable.
Reviewer 2 Report
Dear Authors,
your paper entitled "Prenatal serine biosynthesis deficits underlie the mitochondrial dysfunction associated with the autism-linked FMR1 premutation" by Nolin et al. is well-written and original, adding new insights in the field of the mitochondrial dysfunction associated with FMR1 premutation disorders. I suggest to change title because in prenatal diagnosis is difficult to predict future phenotypes, such as autism (more common in FXS children) of premutated male fetuses, that commonly are unaffected with an increased risk of developing FXTAS.
Minor revisions.
- there are few typos to check (i.e. in Figure 4 is lacking of "A", and "B"; line 135 "unsurpervised");
- in line 43 the authors cited that some mitochondrial dysfunctions were also confirmed by other groups, but they cited their own works. Some others could be those of Hukema et al., 2014; Alvarez-Mora et al., 2017; Nobile et al., 2020, that may be cited;
- in table 1 most of the CGG repeats seem stable through generations, but few others seem reversion of the CGG size. If the authors have the data of the interspersed AGG interruptions it would be useful to insert it in the table 1, where available. In this table the maternal age could be inserted or in the results section the mean±SD of the maternal age;
- the lack of a real control population could also be discussed.
Author Response
Q1. Your paper entitled "Prenatal serine biosynthesis deficits underlie the mitochondrial dysfunction associated with the autism-linked FMR1 premutation" by Nolin et al. is well-written and original, adding new insights in the field of the mitochondrial dysfunction associated with FMR1 premutation disorders. I suggest to change title because in prenatal diagnosis is difficult to predict future phenotypes, such as autism (more common in FXS children) of premutated male fetuses, that commonly are unaffected with an increased risk of developing FXTAS.
A1. We thank the reviewer for his/her positive comments and feedback. We have modified the title for clarity as suggested. Now it reads “Deficits in prenatal serine biosynthesis underlie the mitochondrial dysfunction in the premutations of the autism-linked gene FMR1”.
Q2. There are few typos to check (i.e. in Figure 4 is lacking of "A", and "B"; line 135 "unsurpervised")
A2. We apologize for the oversight. We have labeled the panels in Figure 4.
The word ‘unsupervised’ refers to the type of approach applied to the omics data analysis. Within artificial intelligence (AI) and machine learning, there are two basic approaches: supervised and unsupervised learning. The main difference is one uses labeled data to help predict outcomes, while the other does not. For the first approach (supervised), classification problems (in our case separating the two diagnostic groups) use an algorithm to accurately assign test data into specific categories (i.e., premutation and noncarriers). Linear classifiers, support vector machines, decision trees and random forest are all common types of classification algorithms. In our case, we used PLS-DA. For the second approach, (unsupervised learning) the learning model uses machine learning algorithms to analyze and cluster unlabeled data sets. These algorithms discover hidden patterns in data without the need for human intervention (hence, they are “unsupervised”). We used unsupervised learning models for clustering (visualized as heat maps). Clustering is a data mining technique for grouping unlabeled data based on their similarities or differences. For example, for the heat maps, we used hierarchical clustering, also known as hierarchical cluster analysis (HCA), which is an unsupervised clustering algorithm that can be categorized in two ways; they can be agglomerative or divisive. Agglomerative clustering is considered a “bottoms-up approach.” Its data points are isolated as separate groupings initially, and then they are merged iteratively on the basis of similarity until one cluster has been achieved. Four different methods are commonly used to measure similarity, from which we used the Ward’s linkage. This method states that the distance between two clusters is defined by the increase in the sum of squared after the clusters are merged. This distance was calculated with the Euclidean algorithm as it is the most common metric used to calculate these parameters.
To facilitate the reading of this part, where it read “supervised” or “unsupervised” we added “supervised learning model” or “unsupervised learning model”.
Q3. In line 43 the authors cited that some mitochondrial dysfunctions were also confirmed by other groups, but they cited their own works. Some others could be those of Hukema et al., 2014; Alvarez-Mora et al., 2017; Nobile et al., 2020, that may be cited
A3. We have added the suggested references even though Hukema et al’s study was performed with a mouse model of the premutation with no mitochondrial dysfunction (despite the title), whereas Alvarez-Mora et al’s study was performed in a small set of skin fibroblasts and lymphocytes from noncarriers and FXTAS-affected carriers (<10), and that of Nobile et al’s with <10 samples from FXS and premutation.
Q4. In table 1 most of the CGG repeats seem stable through generations, but few others seem reversion of the CGG size. If the authors have the data of the interspersed AGG interruptions it would be useful to insert it in the table 1, where available.
A4. We have added information on the AGG interruptions in Table 1 (under Fetal CGG repeat structure) and corresponding text and statistics.
Q5. In this table the maternal age could be inserted or in the results section the mean±SD of the maternal age
A5. We have added the maternal age at the time of amniocentesis (Table 1), text and statistics.
Furthermore, in Table 1 we have now marked 2 sets of twins with >80% likelihood of dizygosity (also shown in the new Supplementary Dataset 2). As mentioned in the revised Discussion, analysis of the data by principal component analysis supported the non-concordance of the biological outcomes between dizygotic twins.
To increase our confidence in the results we re-analyzed the data excluding the two sets of dizygotic twins, founding no differences in the conclusions.
Q6. the lack of a real control population could also be discussed.
A6. To do this comparison as it is suggested, then the study question would have been entirely different. If the comparison were between noncarrier mothers and carrier mothers pregnant with noncarrier (noncarrier and carrier mothers) and carrier (carrier mothers) fetuses, then the study question would have been “what role plays the maternal genetic background in the metabolism of the fetus across these 3 groups?”. Our question was, once the genetic background of the mother is taken out of the equation (as they were selected by having similar age, similar CGG repeats, gestational week, and all pregnant with male fetuses), are there any differences in the metabolism of noncarrier vs. carrier fetuses? Thus, to answer our study question, there was no need for a control (noncarrier) population. Future studies may include this group.
To compromise with the reviewer’s comments on adding a limitation to the study, we have added a paragraph before section 3.
Round 2
Reviewer 1 Report
I think the edits and explanations have helped the readability of the paper. I thank the authors for addressing the questions. I accept the authors comments.
Q8 - severity of metabolic block and CGG size - I was thinking about the level of expansion in the fetus rather than the mother as you indicate you are measuring fetal rather than maternal contamination.
A minor query now figure 2B is adjusted - the metabolic impact of CGG size in the amniotic fluid on a larger sample size seems to indicate non carriers have elevated succinate as compared to carriers - this is different to your preposed pathway block but seems to be supported by the cord fibroblasts studies. Can you explain the discrepancy and is the data still consistant. (could the authors consider)
Author Response
Response to reviewer #2:
Q1. I think the edits and explanations have helped the readability of the paper. I thank the authors for addressing the questions. I accept the authors comments.
A1. Thank you very much for your comment. Essentially, we revised the whole paper to make it more suitable for audiences in our and other fields.
Q8 - severity of metabolic block and CGG size - I was thinking about the level of expansion in the fetus rather than the mother as you indicate you are measuring fetal rather than maternal contamination.
A8. We completely agree with the reviewer. This is a follow-up that we were considering doing it with a larger sample size. To evaluate the severity of the block (as the reviewer is indicating) more functional studies are required by using (for example) amniocytes and determine the flow of the metabolites by using labeled compounds. In addition, a larger sample size will allow us to confirm our model in the context of CGG repeats. This relatively small sample size (n=22), spanning from 55 to 157 (mean and SD = 69 ± 2) but with a median of 63, results in a distribution of CGG repeats that are essentially clustered around this last value. We were extremely cautious at presenting our findings by not overstretching our analysis which could be criticized or hampered by the relatively small sample size and by the uneven distribution of CGG repeats.
Q3. A minor query now figure 2B is adjusted - the metabolic impact of CGG size in the amniotic fluid on a larger sample size seems to indicate non carriers have elevated succinate as compared to carriers - this is different to your preposed pathway block but seems to be supported by the cord fibroblasts studies. Can you explain the discrepancy and is the data still consistant. (could the authors consider)
A3. The query was truncated so we will try our best to answer it. To reply to this comment, we need to review the data presented in the study and consider the biochemistry underlying our findings.
First, when the disease-gene pathway analysis was run with the joint input of metabolites and proteins obtained with AF, the “condition” in the PM was mainly associated with AKGDH deficiency (under lipoyltransferase 1, dihydrolipoamide dehydrogenase deficiency, and AKGDH deficiencies, rows 3, 5 and 8 in Table S3; see also Figure S8 with lipoyltransferase 1 deficiency) as well as with deficits in the Ser biosynthetic pathway (last 3 rows in bold Table S3); with the PLS-DA analysis of metabolites (Figure S2 panel D), the main discriminating metabolite was AKG and, as expected, other TCA intermediates (malate, fumarate, succinate); succinate concentrations are not only driven by the activity of the TCA cycle (meaning AKG-derived succinate) but also by other anaplerotic reactions (e.g., catabolism of fatty acids with uneven number of carbons through the formation of propionate; GABA metabolism). In support of this premise, the correlation coefficient for AKG with succinate was lower (and less significant) than that with malate or fumarate (Figure S5); as the TCA is a cycle, it is important to be cognizant of the ratios among metabolites and not their absolute concentration (see with UFC succinate levels in carriers without Gln; Table 2).
Finally, we are aware that the metabolic changes obtained with different cell types or tissues (in our case AF, UFC, PBMCs, skin fibroblasts) may present different degree of severity as it is usually seen with mitochondrial disorders complicated by the phenotypic threshold effect [1-3].
A text clarifying these issues has been added under lines 402-421 (in grey).
- Mitochondrial Medicine Society's Committee on, D.; Haas, R. H.; Parikh, S.; Falk, M. J.; Saneto, R. P.; Wolf, N. I.; Darin, N.; Wong, L. J.; Cohen, B. H.; Naviaux, R. K., The in-depth evaluation of suspected mitochondrial disease. Mol. Genet. Metab. 2008, 94, 16-37, doi:10.1016/j.ymgme.2007.11.018
- Bernier, F. P.; Boneh, A.; Dennett, X.; Chow, C. W.; Cleary, M. A.; Thorburn, D. R., Diagnostic criteria for respiratory chain disorders in adults and children. Neurology 2002, 59, 1406-11, doi:10.1212/01.wnl.0000033795.17156.00
- Rossignol, R.; Faustin, B.; Rocher, C.; Malgat, M.; Mazat, J. P.; Letellier, T., Mitochondrial threshold effects. Biochem. J. 2003, 370, 751-62, doi:10.1042/BJ20021594